# Antifungal Activity of 1,4-Dialkoxynaphthalen-2-Acyl Imidazolium Salts by Inducing Apoptosis of Pathogenic *Candida* spp.

**DOI:** 10.3390/pharmaceutics13030312

**Published:** 2021-02-27

**Authors:** Jisue Lee, Jae-Goo Kim, Haena Lee, Tae Hoon Lee, Ki-Young Kim, Hakwon Kim

**Affiliations:** 1Department of Applied Chemistry, Global Center for Pharmaceutical Ingredient Materials, Kyung Hee University, Seocheon, Giheung, Yongin, Gyeonggi-do 1732, Korea; cwltnc7552@naver.com (J.L.); dlgosk97@naver.com (H.L.); thlee@khu.ac.kr (T.H.L.); 2Graduate School of Biotechnology, Kyung Hee University, Seocheon, Giheung, Yongin, Gyeonggi-do 1732, Korea; zxcv913@naver.com

**Keywords:** 1,4-Dialkoxynaphthalen-2-acyl imidazolium salts, *Candida* sp., antifungal agent, apoptosis

## Abstract

Even though *Candida* spp. are staying commonly on human skin, it is also an opportunistic pathogenic fungus that can cause candidiasis. The emergence of resistant *Candida* strains and the toxicity of antifungal agents have encouraged the development of new classes of potent antifungal agents. Novel naphthalen-2-acyl imidazolium salts (NAIMSs), especially 1,4-dialkoxy-NAIMS from 1,4-dihydroxynaphthalene, were prepared and evaluated for antifungal activity. Those derivatives showed prominent anti-*Candida* activity with a minimum inhibitory concentration (MIC) of 3.125 to 6.26 μg/mL in 24 h based on microdilution antifungal susceptibility test. Among the tested compounds, NAIMS **7c** showed strongest antifungal activity with 3.125 μg/mL MIC value compared with miconazole which showed 12.5 μg/mL MIC value against *Candida* spp., and more importantly >100 μg/mL MIC value against *C. auris*. The production of reactive oxygen species (ROS) was increased and JC-1 staining showed the loss of mitochondrial membrane potential in *C. albicans* by treatment with NAIMS **7c**. The increased release of ultraviolet (UV) absorbing materials suggested that NAIMS **7c** could cause cell busting. The expression of apoptosis-related genes was induced in *C. albicans* by NAIMS **7c** treatment. Taken together, the synthetic NAIMSs are of high interest as novel antifungal agents given further in vivo examination.

## 1. Introduction

Infections by invasive fungal pathogens result from immunosuppression, long-term broad-spectrum antimicrobials, endocrinopathies, organ transplantation and use of indwelling catheters [1,2]. *Candida* spp. is a critical invasive fungal pathogen causing disease in humans, normally responsible for 90% of mucosal infections and 60% of candidiasis episodes [3]. Although various compounds are currently used to control *Candida* infectious diseases, including well-known azoles such as fluconazole, miconazole and others, the mortality of patients with *Candida* infection is above 15% [4,5]. Antibiotic-resistant *Candida* spp. have also arisen. The drugs currently used against fungal pathogens have limitations because of their toxicity [6,7]. For instance, Acetaminophen (APAP), amphotericin B deoxycholate (DAMB) and the triazoles may cause hepatic toxicity [8,9]. Therefore, there is an urgent need for a new drug to treat *Candida* infection.

Imidazolium salts (IMS) have been reported to exhibit fungicidal activity [10]. Due to its ionicity, IMS provides properties that are unusual and highly interesting for pharmaceutical formulation including potential efficacy against some bacteria and fungi [11]. In addition, tuning the toxicity of IMSs as antitumor agents has attracted much attention [12]. Considering these and our previous results of antifungal compound study, we decided to explore in further detail the potential activity of a new hybrid compound formed by attaching an imidazole moiety to 1,4-dialkoxynaphthalen-2-acyl compound to enhance antifungal activity.

Alagebrium, known as an advanced glycation end-products (AGEs) breaker that reverse one of the main mechanisms of ageing, has a structure of phenacyl thiazolium salt [13]. The phenacyl moiety is known to play an important role in biological activity [14,15]. Therefore, we were interested in the napththalenacyl moiety, similar to the phenacyl moiety, as a pharmacophore of antifungal agents. The 1,4-dialkoxy naphthalenacyl compound, derived from 1,4-naphthoquinone, became of particular interest (Figure 1).

Induced endogenous fungal apoptotic responses could provide a basis for antifungal therapies. Environmental stress (acetic acid and hydrogen peroxide) and an antifungal agent (amphotericin B, hibicuslide C and coumarin) have been known to induce apoptosis in *C. albicans* [16,17,18]. Apoptosis is a kind of programmed cell death. Multicellular organisms and even single-celled organisms, such as yeast, can exhibit many features of apoptosis, including DNA fragmentation, reactive oxygen species (ROS) production and the loss of mitochondrial membrane potential [19,20,21].

With the above considerations, a series of novel IMSs linked to a 2-acetyl-1,4-dialkoxynaphthalene moiety were efficiently synthesized through pharmacophore-hybridization strategy (Figure 1) to find promising drugs for dealing with *Candida* spp. infection. Through microdilution antifungal susceptibility, NAIMS **7c** showed the highest antifungal activity among them by inducing *Candida* apoptosis and cell bursting.

## 2. Materials and Methods

### 2.1. General Remarks

The reactions were monitored by thin-layer chromatography (TLC) on Merck Silica gel 60F254. Column chromatography was performed on Merck silica gel 200–300 mesh. Melting points were determined on the melting point apparatus electrothermal A9100X1 and were uncorrected. We recorded ^1^H NMR (300 MHz) and ^13^C NMR (75 MHz) spectra on a JEOL FT-NMR spectrometer (Tokyo, Japan), respectively. Spectra are referenced relative to the chemical shift of tetramethylsilane (TMS). High-resolution mass spectra were obtained with a JEOL JMS-700 mass spectrometer. All solvents and reagents were commercially available from Acros Organics (Brookline, MA, USA), Aldrich (St. Louis, MO, USA) and TCI (Tokyo, Japan) and were used as received. The chemicals 1-Methylimidazole (**6a**) from Sigma-Aldrich (St. Louis, MO, USA) and 1-benzylimidazole (**6b**) from Aldrich are commercially available. We readily obtained 1,4-Diacetoxynaphthalene (**1a**), 1,4-diacetoxy-5-methoxynaphthalene (**1b**), 1,4-diisoamyloxynaphthalene (8), 4-acetoxy-2-acetyl-1-isoamyloxynaphthalene (9**′**) and 2-acetyl-1,4-diisoamyloxynaphthalene (9) from 1,4-naphthoquinone or 5-hydroxy-1,4-naphthoquinone [22]. We prepared 2-Bromoacetylnaphthalene and 2-bromoacetyl-l-methoxynaphthalene from 2-acetylnaphthalene and 2-acetyl-1-methoxynapthalene, respectively, according to the α-bromination procedure.

### 2.2. Synthetic Procedures and Analytical Data

#### 2.2.1. Synthesis of 3-acetyl-4-hydroxynaphthalen-1-yl acetate (**2a**)

1,4-Diacetoxynaphthalene 1a (5 g, 20.4 mmol) was dissolved in boron trifluoride acetic acid complex (20 mL, 144 mmol) and heated under reflux for 1 h. Then, it was cooled to room temperature, quenched with water, extracted by ethyl acetate and washed with water and brine. The combined organic layer was dried over anhydrous magnesium sulfate, filtered and concentrated in vacuo. The residue was purified by flash column chromatography to give the compound **2a** as a pale-yellow solid. (4.9 g, 99%): ^1^H NMR (300 MHz, CDCl_3_) δ ppm 13.92 (s, 1H), 8.44 (d, 1H, J = 8.2 Hz), 7.71 (d, 1H, J = 8.0 Hz), 7.65–7.60 (m, 1H), 7.55–7.50 (m, 1H), 7.35 (s, 1H), 2.59 (s, 3H), 2.42 (s, 3H); ^13^C NMR (75 MHz, CDCl_3_) δ ppm 203.6, 169.6, 160.3, 137.6, 130.8, 130.4, 126.4, 125.8, 124.7, 120.9, 116.3, 111.8, 26.7, 20.7.

#### 2.2.2. Synthesis of 3-acetyl-4-hydroxy-5-methoxynaphthalen-1-yl acetate (**2b**)

Following the procedure described above for the preparation of **2a**, compound **2b** was obtained from 5-methoxy-1,4-diacetoxynaphthalene (**1b**; 0.11 g, 0.4 mmol) as a pale-yellow solid. (0.1 g, 92%): ^1^H NMR (300 MHz, CDCl_3_) δ ppm 14.33 (br s, 1H), 7.58 (dd, 1H, *J* = 8.0 Hz, 8.2 Hz), 7.46 (s, 1H), 7.34 (d, 1H, *J* = 8.2 Hz), 6.95 (d, 1H, *J* = 8.0 Hz), 4.06 (s, 3H), 2.68 (s, 3H), 2.45 (s, 3H).; ^13^C NMR (75 MHz, CDCl_3_) δ ppm 202.5, 169.7, 162.1, 159.7, 137.2, 133.3, 131.0, 117.7, 116.7, 113.4, 112.9, 107.0, 56.2, 27.7, 20.8.

#### 2.2.3. Synthesis of 3-acetyl-4-alkyloxynaphthalen-1-yl acetate (**3a**–**3d**)

##### 3-Aetyl-4-methoxynaphthalen-1-yl acetate (**3a**)

Compound **2a** (0.49 g, 2 mmol) was dissolved in dimethylformamide (DMF, 4 mL). Then, iodomethane (0.19 mL, 3 mmol) and cesium carbonate (0.98 g, 3 mmol) were added to the solution. The mixture was heated under reflux for 1.5 h and was cooled to room temperature. It was quenched with water, extracted by ethyl acetate, and washed with water and brine. The combined organic layer was dried over anhydrous magnesium sulfate, filtered and concentrated in vacuo. The residue was purified by flash column chromatography to afford the compound **3a** as a yellow solid. (0.41 g, 80%): ^1^H NMR (300 MHz, CDCl_3_) δ ppm 8.25–8.21 (m, 1H), 7.86–7.81 (m, 1H), 7.67–7.58 (m, 2H), 7.54 (s, 1H), 4.01 (s, 3H), 2.78 (s, 3H), 2.46 (s, 3H). ^13^C NMR (75 MHz, CDCl_3_) δ ppm 198.0, 169.1, 155.3, 142.5, 129.8, 128.7, 128.5, 126.8, 123.5, 121.4, 117.2, 63.6, 30.3, 20.4.

##### 3-Aetyl-4-(isoamyloxy)naphthalen-1-yl acetate (**3b**)

Following the procedure described above for the preparation of **3a**, compound **3b** was obtained from 2a (0.49 g, 2 mmol) and 1-bromo-3-methyl butane (0.36 mL, 3 mmol) as a yellow oil. (0.44 g, 71%): ^1^H NMR (300 MHz, CDCl_3_) δ ppm 8.23–8.18 (m, 1H), 7.84–7.79 (m, 1H), 7.66–7.56 (m, 2H), 7.49 (s, 1H), 4.07-4.01 (m, 2H), 2.76 (s, 3H), 2.45 (s, 3H), 1.89–1.82 (m, 3H), 1.05 (d, 6H, J = 9.0 Hz). ^13^C NMR (75 MHz, CDCl_3_) δ ppm 199.2, 169.4, 154.6, 142.5, 129.9, 129.4, 128.7, 127.7, 127.0, 123.7, 121.6, 117.5, 76.2, 39.0, 30.5, 25.0, 22.6, 20.8.

##### 3-Acetyl-4-isopropoxynaphthalen-1-yl acetate (**3c**)

Following the procedure described above for the preparation of **3a**, compound **3c** was obtained from **2a** (0.49 g, 2 mmol) and 2-bromopropane (0.28 mL, 3 mmol) as a yellow oil. (0.36 g, 63%): ^1^H NMR (300 MHz, CDCl_3_) δ ppm 8.23–8.18 (m, 1H), 7.83–7.80 (m, 1H), 7.74–7.54 (m, 2H), 7.41 (s, 1H), 4.39–4.27 (m, 1H), 2.75 (s, 3H), 2.45 (s, 3H), 1.34 (d, 6H, J = 6.0 Hz). ^13^C NMR (75 MHz, CDCl_3_) δ ppm 200.8, 169.3, 152.2, 142.3, 130.0, 129.5, 129.1, 128.4, 126.6, 124.1, 121.4, 117.2, 79.5, 60.2, 30.4, 22.2, 20.7.

##### 3-Acetyl-4-(isoamyloxy)-5-methoxynaphthalen-1-yl acetate (**3d**)

Following the procedure described above for the preparation of **3a**, compound **3d** was obtained from **2b** (0.1 g, 0.36 mmol) as a yellow solid. (0.98 g, 82%): ^1^H NMR (300 MHz, CDCl_3_) δ ppm 7.51 (dd, 1H, J = 8.2 Hz, 7.6 Hz), 7.45 (s, 1H), 7.41 (d, 1H, J = 8.4 Hz), 6.95 (d, 1H, J = 7.6 Hz), 4.03 (s, 3H), 3.90 (t, 2H, J = 7.2 Hz), 2.76 (s, 3H), 2.44 (s, 3H), 1.87–1.75 (m, 3H), 0.97 (d, 6H, J = 6.2 Hz). ^13^C NMR (75 MHz, CDCl_3_) δ ppm 200.5, 169.3, 157.2, 154.9, 142.4, 132.2, 129.7, 128.9, 120.8, 118.3, 114.0, 106.9, 56.1, 56.0, 38.9, 31.3, 29.6, 25.1, 22.7, 20.9.

#### 2.2.4. General Synthesis of 2-acetyl-1,4-dialkoxynaphthalene (**4**)

##### 2-Acetyl-1,4-dimethoxynaphthalene (**4a**; CAS Registry Number 65131-13-7)

**3a** (0.29 g, 1.12 mmol) was dissolved in methanol (6 mL) and was cooled to 0 °C. Then 1 wt % potassium hydroxide solution in methanol (5.4 mL) was added. After stirring for 2 h, the mixture was neutralized by adding Amberlite IR-120(H) and stirred for additional 15 m. Amberlite IR-120(H) was removed by filtration and the filtrate was concentrated in vacuo. The resulting residue was dissolved in DMF (2 mL), and iodomethane (0.11 mL, 1.68 mmol) and cesium carbonate (0.5 g, 1.68 mmol) were added. The mixture was heated under reflux. After monitoring the reaction complete by TLC, it was cooled to room temperature, extracted with dichloromethane (DCM), washed with water and dried over anhydrous magnesium sulfate. It was concentrated in vacuo and purified by flash column chromatography to give the product **4a** as a yellow oil. (0.26 g, quantitative yield)- ^1^H NMR (300 MHz, CDCl_3_) δ ppm 8.28–8.23 (m, 1H), 8.19–8.15 (m, 1H), 7.63–7.57 (m, 2H), 7.08 (s, 1H), 4.01 (s, 3H), 3.96 (s, 3H), 2.81 (s, 3H). ^13^C NMR (75 MHz, CDCl3) δ ppm.

##### 2-Acetyl-1,4-diisoamyloxynaphthalene (**4b**)

Following the procedure described above for the preparation of **4a**, compound **4b** was obtained from **3b** (1 g, 3.3 mmol) and 1-bromo-3-methylbutane (0.6 mL, 5 mmol) as a yellow oil. (0.47 g, 42%): ^1^H NMR (300 MHz, CDCl_3_) δ ppm 8.29–8.24 (m, 1H), 8.16–8.10 (m, 1H), 7.60–7.55 (m, 2H), 7.01 (s, 1H), 4.17 (t, 2H, J = 6.3 Hz), 3.97 (t, 2H, J = 6.6 Hz), 2.78 (s, 1H), 1.00 (d, 6H, J = 6.3 Hz), 0.98 (d, 6H, J = 5.4 Hz). ^13^C NMR (75 MHz, CDCl_3_) δ ppm 200.5, 151.0, 150.3, 128.9, 127.7, 127.4, 126.8, 122.9, 122.5, 102.6, 75.8, 66.6, 39.1, 37.8, 30.7, 25.1, 24.9, 22.6, 22.5.

##### 2-Acetyl-1,4-diisopropoxynaphthalene (**4c**)

Following the procedure described above for the preparation of **4a**, compound **4c** was obtained from **3c** (0.12 g, 0.42 mmol) as a yellow oil. (0.1 g, 83%): ^1^H NMR (300 MHz, CDCl_3_) δ ppm 8.28–8.25 (m, 1H), 8.15–8.11 (m, 1H), 7.60–7.53 (m, 2H), 6.95 (s, 1H), 4.80–4.72 (m, 1H), 4.31–4.25 (m, 1H), 2.76 (s, 3H), 1.45 (s, 3H), 1.43 (s, 3H), 1.31(d, 6H, J = 6.0 Hz).; ^13^C NMR (75 MHz, CDCl_3_) δ ppm 202.5, 149.6, 147.7, 130.0, 129.5, 129.4, 127.2, 126.5, 123.5, 122.6, 104.4, 78.8, 70.5, 30.8, 22.3, 22.1.

##### 2-Acetyl-1,4-diisoamyloxy-8-methoxynaphthalene (**4d**)

Following the procedure described above for the preparation of **4a**, compound **4d** was obtained from **3d** (0.09 g, 0.27 mmol) as a yellow solid. (0.09 g, 77%): ^1^H NMR (300 MHz, CDCl_3_) δ ppm 7.90 (d, 1H, J = 8.2 Hz). 7.46 (dd, 1H, J = 8.2 Hz, 7.8 Hz), 7.00 (s, 1H), 6.95 (d, 1H, J = 7.6 Hz), 4.15 (t, 2H, J = 6.4 Hz), 4.01 (s, 3H), 3.85 (t, 2H, J = 7.1 Hz), 2.78 (s, 3H), 1.98–1.75 (m, 6H), 1.00 (d, 6H, J = 6.4 Hz), 0.96 (d, 6H, J = 6.2 Hz).; ^13^C NMR (75 MHz, CDCl_3_) δ ppm 202.1, 156.7, 150.7, 150.3, 131.1, 129.7, 127.6, 120.3, 114.9, 107.0, 103.7, 76.4, 66.7, 55.9, 38.9, 37.9, 31.5, 25.2, 25.1, 22.7, 22.6.

#### 2.2.5. General Synthesis of naphthalenacyl bromide (**5a**–**5e**)

##### 2-Bromoacetyl-1,4-dimethoxynaphthalne (**5a**)

**4a** (0.2 g, 0.86 mmol) was dissolved in DCM (10 mL). Tetrabutylammonium tribromide (TBA-Br_3_; 0.36 g, 1.12 mmol) was added and the resulting solution was stirred under argon atmosphere for 3 h. It was quenched with water, extracted with DCM and washed with water, 1M sodium bicarbonate solution and brine, successively. The combined organic phase was dried over anhydrous sodium sulfate and concentrated in vacuo. It was purified by flash column chromatography to give the compound **5a** as a yellow solid. (0.18 g, 67%) [23]: ^1^H NMR (300 MHz, CDCl_3_) δ ppm 8.30 (d, 1H, J = 8.3 Hz), 8.20 (d, 1H, J = 7.9 Hz), 7.74 (t, 1H, J = 7.7 Hz), 7.65 (t, 1H, J = 7.3Hz), 6.82 (s, 1H), 4.82 (s, 2H), 3.99 (s, 3H).^13^C NMR (75 MHz, (CD_3_)_2_SO) δ ppm 192.8, 152.0, 151.8, 129.3, 128.1, 127.2, 124.4, 123.2, 122.6, 102.1, 64.2, 55.7, 36.4, 36.4.

##### 2-Bromoacetyl-1,4-diisoamyloxynaphthalene (**5b**)

Following the procedure described above for the preparation of **5a**, compound **5b** was obtained from **4b** (3.76 g, 11 mmol) as a yellow oil. (2.8 g, 62%): ^1^H NMR (300 MHz, CDCl_3_) δ ppm 8.31-8.27 (m, 1H), 8.12-8.09 (m, 1H), 7.63-7.57 (m, 2H), 6.99 (s, 1H), 4.79 (s, 2H), 4.19-4.15 (m, 2H), 4.02-3.97 (m, 2H), 1.99–1.88 (m, 2H), 1.86–1.78 (m, 4H), 1.01 (d, 6H, J = 6.6 Hz), 1.99 (d, 6H, J = 6.3 Hz). ^13^C NMR (75 MHz, CDCl_3_) δ ppm 193.9, 151.4, 150.4, 129.4, 128.5, 128.0, 127.2, 125.1, 123.0, 122.7, 102.7, 76.4, 66.8, 39.0, 37.8, 36.2, 25.2, 25.0, 22.6, 22.6.

##### 2-Bromoacetyl-1,4-diisopropoxynaphthalene (**5c**)

Following the procedure described above for the preparation of **5a**, compound **5c** was obtained from **4c** (0.17 g, 0.63 mmol) as a yellow oil. (0.13 g, 56%): ^1^H NMR (300 MHz, CDCl_3_) δ ppm 8.30–8.27 (m, 1H), 8.10–8.08 (m, 1H), 7.60–7.55 (m, 2H), 6.92 (s, 1H), 4.79 (s, 2H), 4.79–4.76 (m, 1H), 4.32–4.26 (m, 1H), 1.55–1.43 (m, 6H), 1.33–1.30 (m, 6H). ^13^C NMR (75 MHz, CDCl_3_) δ ppm 195.7, 150.0, 147.7, 129.8, 129.4, 127.6, 126.9, 123.4, 122.9, 104.3, 100.5, 70.6, 36.0, 22.3, 22.0.

##### 2-Bromoacetyl-1,4-diisoamyloxy-8-methoxynaphthalene (**5d**)

Following the procedure described above for the preparation of 5a, compound **5d** was obtained from **4d** (0.67 g, 1.8 mmol) as a yellow solid. (0.53 g, 65%): ^1^H NMR (300 MHz, CDCl_3_) δ ppm 7.91 (d, 1H, J = 8.4 Hz), 7.50 (dd, 1H, J = 7.8 Hz, 8.4 Hz), 6.97 (d, 1H, J = 7.8 Hz), 6.96 (s, 1H), 4.81 (s, 2H), 4.15 (t, 2H, J = 6.2 Hz), 4.02 (s, 3H), 3.86 (t, 2H, J = 7.1 Hz), 1.98–1.72 (m, 6H), 1.00 (d, 6H, J = 6.5 Hz), 0.96 (d, 6H, J = 6.2 Hz). ^13^C NMR (CDCl_3_, 75 MHz) δ ppm 195.1, 156.7, 151.1, 150.4, 131.5, 128.2, 127.0, 120.0, 115.1, 107.4, 103.9, 76.8, 66.9, 56.0, 38.9, 37.9, 37.1, 25.2, 25.1, 22.7, 22.6.

##### 2-Bromoacetyl-1-methoxynaphthalene (**5e**)

Following the procedure described above for the preparation of **5a**, compound **5e** was obtained from 2-acetyl-1-methoxynaphthalene (0.5 g, 2.5 mmol) as a white solid. (0.49 g, 70%): ^1^H NMR (300 MHz, CDCl_3_) δ ppm 8.21–8.18 (m, 1H), 7.88–7.85 (m, 1H), 7.74 (d, 1H, J = 8.6 Hz), 7.65 (d, 1H, J = 8.4 Hz), 7.62–7.56 (m, 2H), 4.74 (s, 2H), 4.03 (s, 3H).; ^13^C NMR (75 MHz, CDCl_3_) δ ppm 193.0, 157.7, 137.2, 128.7, 128.2, 127.5, 126.8, 125.6, 124.9, 124.6, 123.3, 64.2, 36.3.

#### 2.2.6. General Synthesis of 1-substituted benzylimidazoles (**6c** and **6d**)

##### 1-(4-Methoxybenzyl)-1H-imidazole (**6c**)

Imidazole (0.25 g, 3.68 mmol) and potassium carbonate (0.5 g, 3.68 mmol) were dissolved in acetonitrile (15 mL). Then 4-methoxybenzyl chloride (0.5 mL, 3.68 mmol) was added and the reaction mixture was stirred for 12 h at room temperature. After completion of the reaction, it was concentrated under reduced pressure and purified by flash column chromatography to obtain compound **6c** in the form of an ivory solid. (0.4 g, 60%) [24]: ^1^H NMR (300 MHz, CDCl_3_) δ ppm 7.52 (s, 1H), 7.12–7.07 (m, 3H), 6.89–6.86 (m, 3H), 5.04 (s, 2H), 3.80 (s, 3H). ^13^C NMR (75 MHz, CDCl_3_) δ ppm 159.4, 137.1, 129.6, 128.7, 128.0, 119.0, 114.2, 55.2, 50.1.

##### 1-(4-Nitrobenzyl)-1H-imidazole (**6d**)

Following the procedure described above for the preparation of **6c**, compound **6d** was obtained from 4-nitrobenzyl bromide (0.5 g, 2.31 mmol) as a yellow solid. (0.46 g, >98%): ^1^H NMR (300 MHz, CDCl_3_) δ ppm 8.22 (d, 2H, J = 8.6 Hz), 7.59 (s, 1H), 7.28 (d, 2H, J = 6.5 Hz), 7.15 (s, 1H), 6.92 (s, 1H), 5.26 (s, 2H).; ^13^C NMR (75 MHz, CDCl_3_) δ ppm 147.3, 143.4, 137.3, 129.9, 127.5, 123.8, 119.1, 49.5.

#### 2.2.7. General Synthesis of NAIMSs (**7a**–**7i**, **10**, **11**)

##### 3-(2-(1,4-Dimethoxynaphthalen-2-yl)-2-oxoethyl)-1-methyl-1H-imidazol-3-ium bromide (NAIMS **7a**)

Compound **5a** (0.05g, 0.16mmol) and 1-methylimidazole (6a; 0.025 mL, 0.32 mmol) were dissolved in acetonitrile (2 mL). The mixture was heated under reflux for 1 d. It was concentrated under reduced pressure and recrystallized by acetonitrile and ether to give the compound **7a** as an ivory solid. (0.061 g, 98%) [25]: ^1^H NMR (300 MHz, (CD_3_)_2_SO) δ ppm 9.10 (s, 1H), 8.26–8.22 (m, 2H), 7.78–7.76 (m, 4H), 7.19 (s, 1H), 5.98 (s, 2H), 4.09 (s, 3H), 3.99 (s, 3H), 3.97 (s, 3H). ^13^C NMR (75 MHz, (CD_3_)_2_SO) δ ppm 191.1, 152.8, 151.3, 137.8, 129.1, 128.9, 127.9, 124.1, 123.8, 123.2, 123.1, 122.2, 101.1, 64.1, 58.2, 55.8, 35.9. m.p. 227.6 °C decomposed. HRMS (FAB) m/z Calcd. for C_18_H_19_N_2_O_3_ [M − Br]^+^ 311.1396, found 311.1395.

##### 3-(2-(1,4-Bis(isoamyloxy)naphthalen-2-yl)-2-oxoethyl)-1-methyl-1H-imiazol-3-ium bromide (NAIMS **7b**)

Following the procedure described above for the preparation of **7a**, compound **7b** was obtained from **5b** (0.12 g, 0.28 mmol) and imidazole **6a** (0.045 g, 0.56 mmol) as an ivory solid. (0.1 g, 71%): ^1^H NMR (300 MHz, (CD_3_)_2_SO) δ 9.12 (s, 1H), 8.24 (dd, 1H, J = 3.1, 6.2 Hz), 8.17 (dd, 1H, J = 3.1, 5.8 Hz), 7.79–7.76 (m, 4H), 7.21 (s, 1H), 4.23–4.14 (m, 4H), 3.97 (s, 3H), 1.97–1.85 (m, 4H), 1.81–1.74 (m, 2H), 0.99–0.97 (m, 12H). ^13^C NMR (75 MHz, (CD_3_)_2_SO) δ 191.1, 151.3, 150.6, 137.8, 129.0, 128.2, 127.8, 124.0, 123.6, 123.3, 123.1, 122.3, 102.0, 75.5, 66.5, 57.9, 38.4, 37.2, 35.9, 24.8, 24.6, 22.6, 22.4. m.p. 68.9–70.8 °C. HRMS (FAB) m/z Calcd. for C_26_H_35_N_2_O_3_ [M − Br]^+^ 423.2648, found 423.2644.

##### 1-Benzyl-3-(2-(1,4-bis(isoamyloxy)naphthalen-2-yl)-2-oxoethyl)-1H-imiazol-3-ium bromide (NAIMS **7c**)

Following the procedure described above for the preparation of **7a**, compound **7c** was obtained from **5b** (0.05g, 0.12mmol) and 1-benzylimidazole (**6b**; 0.037 g, 0.24 mmol) to give the compound **7c** as an ivory solid. (0.07 g, >98%): ^1^H NMR (300 MHz, (CD_3_)_2_SO) δ ppm 9.30 (s, 1H), 8.23 (dd, 1H, J = 2.9, 6.0 Hz), 8.17–8.15 (m, 1H), 7.92 (s, 1H), 7.80 (s, 1H), 7.77–7.74 (m, 2H), 7.51–7.43 (m, 5H), 7.21 (s, 1H), 5.98 (s, 2H), 5.58 (s, 2H), 4.22–4.16 (m, 4H), 2.21–1.88 (m, 4H), 1.80–1.76 (m, 2H), 0.99–0.97 (m, 12H). ^13^C NMR (75 MHz, (CD_3_)_2_SO) δ ppm 191.0, 151.3, 150.5, 137.5, 134.8, 129.0, 128.7, 128.2, 128.1, 127.8, 124.4, 123.5, 123.3, 122.2, 122.0, 101.9, 75.5, 66.5, 58.1, 51.9, 38.4, 37.2, 24.7, 24.5, 22.5, 22.4. m.p. 87.7–88.8 °C. HRMS (FAB) m/z Calcd. for C_32_H_39_N_2_O_3_ [M − Br]^+^ 499.2961, found 499.2960.

##### 3-(2-(1,4-Bis(isoamyloxy)naphthalen-2-yl)-2-oxoethyl)-1-(4-methoxybenzyl)-1H-imidazol-3-ium bromide (NAIMS **7d**)

Following the procedure described above for the preparation of **7a**, compound **7d** was obtained from **5b** (0.06 g, 0.14 mmol) and 4-methoxybenzylimidazole (**6c**; 0.05 g, 0.28 mmol) as an ivory solid. (0.045 g, 55%): ^1^H NMR (300 MHz, (CD_3_)_2_SO) δ 9.17 (s, 1H), 8.23 (dd, 1H, J = 3.1, 6.2 Hz), 8.15 (dd, 1H, J = 3.3, 6.2 Hz), 7.86 (s, 1H), 7.77–7.74 (m, 3H), 7.44 (d, 2H, J = 8.4 Hz), 7.18 (s, 1H), 7.02 (d, 2H, J = 8.6 Hz), 5.92 (s, 2H), 5.46 (s, 2H), 4.21–4.12 (m, 4H), 1.90–1.87 (m, 4H), 1.80–1.74 (m, 2H), 0.98 (d, 6H, J = 2.9 Hz), 0.97 (d, 6H, J = 3.6 Hz). ^13^C NMR (75 MHz, (CD_3_)_2_SO) δ 191.0, 159.6, 151.4, 150.5, 137.2, 130.0, 129.0, 128.1, 127.8, 126.5, 124.4, 123.5, 123.3, 122.2, 121.8, 114.4, 101.9, 75.5, 66.5, 58.1, 55.2, 51.5, 38.4, 37.2, 24.7, 24.5, 22.5, 22.4. m.p. 113.3–114.7 °C. HRMS (FAB) m/z Calcd. for C_33_H_41_N_2_O_4_ [M − Br]^+^ 529.3066, found 529.3068.

##### 3-(2-(1,4-Bis(isoamyloxy)naphthalen-2-yl)-2-oxoethyl)-1-(4-nitrobenzyl)-1H-imidazol-3-ium bromide (NAIMS **7e**)

Following the procedure described above for the preparation of **7a**, compound **7e** was obtained from **5b** (0.06 g, 0.14 mmol) and 1-(4-nitrobenzyl)imidazole (**6d**; 0.04 g, 0.21 mmol) as an ivory solid. (0.06 g, 70%): ^1^H NMR (300 MHz, (CD_3_)_2_SO) δ 9.28 (s, 1H), 8.34 (d, 2H, J = 6.7 Hz), 8.23–8.07 (m, 2H), 7.92 (s, 1H), 7.81 (s, 1H), 7.78–7.75 (m, 2H), 7.70 (d, 2H, J = 7.1 Hz), 7.20 (s, 1H), 5.96 (s, 2H), 5.74 (s, 2H), 4.22–4.14 (m, 4H), 1.99–1.91 (m, 4H), 1.79–1.76 (m, 2H), 0.99–0.96 (m, 12H).; ^13^C NMR (75 MHz, (CD_3_)_2_SO) δ 190.9, 151.4, 150.6, 147.6, 142.1, 137.9, 129.4, 129.0, 128.1, 127.8, 124.7, 124.1, 123.5, 123.3, 122.2, 122.1, 101.9, 75.6, 66.5, 58.2, 51.0, 38.4, 37.2, 24.7, 24.5, 22.5, 22.4.; m.p. 160.2–160.6 °C; HRMS (FAB) m/z Calcd. for C_32_H_38_N_3_O_5_ [M − Br]^+^ 544.2811, found 544.2812.

##### 1-Benzyl-3-(2-(1,4-diisopropoxynaphthalen-2-yl)-2-oxoethyl)-1H-imidazol-3-ium bromide (NAIMS **7f**)

Following the procedure described above for the preparation of **7a**, compound **7f** was obtained from **5c** (0.08 g, 0.22 mmol) and 1-benzylimidazole (**6b**; 0.06 g, 0.38 mmol) as an ivory solid. (0.065 g, 59%): ^1^H NMR (300 MHz, CDCl_3_) δ 11.00 (s, 1H), 8.32–8.30 (m, 1H), 8.10–8.07 (m, 1H), 7.63–7.60 (m, 2H), 7.47–7.42 (m, 5H), 7.12–7.10 (m, 2H), 7.04 (s, 1H), 6.08 (s, 2H), 5.61 (s, 2H), 4.81–4.77 (m, 1H), 4.55–4.51 (m, 1H), 1.47–1.45 (m, 12H). ^13^C NMR (75 MHz, CDCl_3_) δ ppm 192.2, 150.2, 149.8, 139.2, 132.5, 130.7, 129.6, 129.5, 129.1, 128.9, 128.4, 127.0, 125.2, 123.8, 123.1, 123.1, 120.8, 103.1, 79.7, 70.8, 58.6, 53.5, 22.4, 22.0. m.p. 191.7–192.5 °C. HRMS (FAB) m/z Calcd. for C_28_H_31_N_3_O_3_ [M − Br]^+^ 443.2335, found 443.2332.

##### 3-(2-(1,4-Bis(isoamyloxy)-8-methoxynaphthalen-2-yl)-2-oxoethyl)-1-methyl-1H-imidazol-3-ium bromide (NAIMS **7g**)

Following the procedure described above for the preparation of **7a**, compound **7g** was obtained from **5d** (0.05 g, 0.11 mmol) and 1-methylimidazole (**6a**; 0.018 g, 0.22 mmol) as an ivory solid. (0.051 g, 88%): ^1^H NMR (300 MHz, (CD_3_)_2_SO) δ ppm 9.06 (s, 1H), 7.81 (d, 1H, J = 8.4 Hz), 7.75 (s, 1H), 7.72 (s, 1H), 7.68–7.62 (m, 1H), 7.21 (d, 1H, J = 8.0 Hz), 7.17 (s, 1H), 5.87 (s, 2H), 4.18–4.13 (m, 2H), 3.99 (s, 3H), 3.95 (s, 3H), 3.95–3.92 (m, 2H), 1.89–1.87 (m, 2H), 1.84–1.72 (m, 4H), 0.96 (d, 6H, J = 5.5 Hz), 0.95 (d, 6H, J = 5.6 Hz). ^13^C NMR (75 MHz, (CD_3_)_2_SO) δ ppm 191.6, 156.9, 152.5, 150.0, 137.8, 131.3, 129.7, 124.3, 123.9, 123.0, 119.3, 114.0, 108.3, 102.7, 75.8, 66.4, 58.2, 56.0, 38.2, 37.2, 35.8, 24.8, 24.7, 22.7, 22.4. m.p. 112.9–114.7 °C. HRMS (FAB) m/z Calcd. for C_27_H_37_N_2_O_4_ [M − Br]^+^ 453.2753, found 453.2752.

##### 1-Benzyl-3-(2-(1,4-bis(isoamyloxy)-8-methoxynaphthalen-2-yl)-2-oxoethyl)-1H-imidazol-3-ium bromide (NAIMS **7h**)

Following the procedure described above for the preparation of **7a**, compound **7h** was obtained from **5d** (0.1 g, 0.22 mmol) and imidazole **6b** (0.05 g, 0.33 mmol) as an ivory solid. (0.1 g, 80%): ^1^H NMR (300 MHz, (CD_3_)_2_SO) δ ppm 9.22 (s, 1H), 7.87 (s, 1H), 7.80 (d, 1H, J = 8.4 Hz), 7.74 (s, 1H), 7.67–7.62 (m, 1H), 7.48–7.42 (m, 5H), 7.20 (d, 1H, J = 7.7 Hz), 7.17 (s, 1H), 5.87 (s, 2H), 5.54 (s, 2H), 4.17–4.12 (m, 2H), 3.99 (s, 3H), 3.95–3.91 (m, 2H), 1.90–1.84 (m, 4H), 1.75–1.73 (m, 2H), 0.96 (d, 6H, J = 6.6 Hz), 0.94 (d, 6H, J = 6.5 Hz). ^13^C NMR (75 MHz, (CD_3_)_2_SO) δ ppm 191.5, 156.9, 152.4, 150.0, 137.5, 134.8, 131.3, 129.6, 129.0, 128.7, 128.2, 124.4, 124.3, 121.9, 119.3, 114.0, 108.2, 102.7, 75.9, 66.4, 58.5, 56.0, 51.9, 38.1, 37.2, 24.8, 24.7, 22.6, 22.3. m.p. 121.3–122.7 °C. HRMS (FAB) m/z Calcd for C_33_H_41_N_2_O_4_ [M − Br]^+^ 529.3066, found 529.3068.

##### 3-(2-(1,4-Bis(isoamyloxy)-8-methoxynaphthalen-2-yl)-2-oxoethyl)-1-(4-nitrobenzyl)-1H-imidazol-3-ium bromide (NAIMS **7i**)

Following the procedure described above for the preparation of **7a**, compound **7i** was obtained from **5d** (0.1 g, 0.22 mmol) and imidazole **6d** (0.067 g, 0.33 mmol) as an ivory solid. (0.12 g, 88%): ^1^H NMR (300 MHz, (CD_3_)_2_SO) δ ppm 9.27 (s, 1H), 8.34 (d, 2H, J = 8.0 Hz), 7.90 (s, 1H), 7.81 (d, 1H, J = 8.4 Hz), 7.79 (s, 1H), 7.69 (d, 2H, J = 8.6 Hz), 7.68–7.62 (m, 1H), 7.21 (d, 1H, J = 8.0 Hz), 7.17 (s, 1H), 5.90 (s, 2H), 5.73 (s, 2H), 4.17–4.13 (m, 2H), 3.99 (s, 3H), 3.94 (t, 2H, J = 7.3 Hz), 1.87–1.83 (m, 4H), 1.77–1.73 (m, 2H), 0.96 (d, 6H, J = 6.3 Hz), 0.94 (d, 6H, J = 6.2 Hz). ^13^C NMR (75 MHz, (CD_3_)_2_SO) δ ppm 191.4, 156.9, 152.5, 150.0, 147.6, 142.1, 137.9, 131.3, 129.7, 129.4, 124.6, 124.2, 124.0, 122.0, 119.3, 114.0, 108.2, 102.7, 75.9, 66.4, 58.5, 56.0, 51.0, 38.2, 37.2, 24.8, 24.7, 22.6, 22.4. m.p. 158.3–159.8 °C. HRMS (FAB) m/z Calcd. for C_33_H_40_N_3_O_6_ [M − Br]^+^ 574.2917, found 574.2916.

##### 1-Benzyl-3-(2-(naphthalen-2-yl)-2-oxoethyl)-1H-imidazol-3-ium bromide (NAIMS **10**)

Following the procedure described above for the preparation of **7a**, compound **10** was obtained from 2-bromoacetylnapththalene (0.1 g, 0.4 mmol) and imidazole **6b** as a white solid. (0.09 g, 56%): ^1^H NMR (300 MHz, (CD_3_)_2_SO) δ ppm 9.24 (s, 1H), 8.79 (s, 1H), 8.19 (d, 1H, J = 8.0 Hz), 8.13 (d, 1H, J = 8.2 Hz), 8.06 (d, 1H, J = 8.2 Hz), 7.91 (s, 1H), 7.76 (s, 1H), 7.75–7.69 (m, 2H), 7.47–7.45 (m, 5H), 6.16 (s, 2H), 5.56 (s, 2H). ^13^C NMR (75 MHz, (CD_3_)_2_SO) δ ppm 191.1, 137.5, 135.4, 134.8, 131.9, 130.9, 130.4, 129.6, 129.3, 129.0, 128.8, 128.7, 128.2, 127.8, 127.4, 124.4, 123.1, 122.2, 55.5, 52.0. m.p. 170.8 °C decomposed. HRMS (FAB) m/z Calcd. for C_22_H_19_N_2_O [M − Br]^+^ 327.1497, found 327.1497.

##### 1-Benzyl-3-(2-(1-methoxynaphthalen-2-yl)-2-oxoethyl)-1H-imidazol-3-ium bromide (NAIMS **11**)

Following the procedure described above for the preparation of **7a**, compound **11** was obtained from 2-bromoacetyl-l-methoxynaphthalene **5e** (0.1 g, 0.36 mmol) and imidazole **6b** as an ivory solid. (0.12 g, 79%): ^1^H NMR (300 MHz, (CD_3_)_2_SO) δ ppm 9.39 (s, 1H), 8.29 (d, 1H, *J =* 7.5 Hz), 8.07 (d, 1H, *J =* 8.4 Hz), 7.95 (s, 1H), 7.91–7.88 (m, 2H), 7.84 (s, 1H), 7.78–7.70 (m, 2H), 7.49–7.40 (m, 5H), 6.04 (s, 2H), 5.61 (s, 2H), 4.16 (s, 3H). ^13^C NMR (75 MHz, (CD_3_)_2_SO) δ ppm 191.1, 158.8, 137.5, 137.2, 134.8, 129.4, 129.0, 128.7, 128.3, 128.2, 127.2, 127.1, 124.6, 124.5, 124.4, 123.6, 123.4, 121.9, 64.2, 58.4, 51.9. m.p. 159.5–159.8 °C. HRMS (FAB) m/z Calcd. for C_23_H_21_N_2_O_2_ [M − Br]^+^ 357.1603, found 357.1604.

### 2.3. Fungal Strains and Culture

*Candida tropicalis* var. *tropicalis* (KCTC17762), *Candida glabrata* (KCTC7219), *Candida tropicalis* (KCTC7212), *Candida albicans* (KCTC7270), *Candida albicans* (KCTC7965) and *Candida auris* (KCTC17810) were purchased from KCTC (Korean Collection for Type Cultures, *Candida parapsilosis* var. *parapsilosis* (KACC45480) and *Candida albicans* (KACC30071) were purchased from KACC (Korean Agricultural Culture Collection, Suwon, Korea). All strains were kept in 20% glycerol at −70 °C. They were cultured in YPD containing yeast extract 10 g/L (BD Difco, Franklin Lakes, NJ, USA), peptone 20 g/L (BD Difco) and 2% D-glucose (*w/v*) (Daejung, Gyeonggi-do, Suwon, Korea) at 30 °C for 24–48 h [26].

### 2.4. Antifungal Susceptibility Microdilution Assay

Microdilution assay was performed based on the description of CLSI document M27-A [26,27,28]. Compounds were prepared in DMSO (Dimethyl sulfoxide; Junsei, Tokyo, Japan) at a concentration of 10 mg/mL as a stock solution. A colony of each strain was inoculated in 3 mL of YPD broth at 30 °C overnight, and the medium was changed to new fresh medium. The strains were adjusted at 3.0 × 10^4^ CFU/mL with the final compound concentrations ranging from 1.5625 to 100 μg/mL. Growth control without treatment and sterilized medium control were included in each experiment and miconazole was used as the positive control [26,29].

### 2.5. Cell Viability Assay

Cell viability was estimated according to previously reported method with slightly modification [30]. A normal cell line, HaCaT (ATCC, VA, USA), was added to a 96-well plate at 1.0 × 10^4^ cells per well and incubated for 24 h. Various concentrations of NAIMS 7c (3.125–100 μg/mL) were added and further incubated for 24 h. MTT (3-(4,5-dimethyl-thiazol-2-yl)-2,5-diphenyltetrazolium bromide, Sigma, St. Louis, MO, USA) in PBS was added into each well, followed by incubation for 3 h at 37 °C. The medium was then removed, and cells were suspended in 100 μL DMSO for 10 m. Viable cells were calculated from optical density (OD_540_) values measured using a microplate reader (BioTek Instruments, Winooski, VT, USA).

### 2.6. Fungal Cell Growth Test

Inoculation and culture conditions in this study were same as those in the above-mentioned antifungal susceptibility microdilution assay. The compound concentrations then ranged from 1.56 to 25 μg/mL. We cultured 96-well plates (SPL, Gyeonggi-do, Korea) at 30 °C and OD_600_ were measured using a microplate reader (BioTek Instruments) at indicated time [31].

### 2.7. ROS Detection

ROS detection cell-based assay kit (Cayman Chemical, Ann Arbor, MI, USA) was used according to manufacturer’s instructions. Cells were incubated with/without antimycin (positive control) or indicated concentrations of NAIMS **7c** for 2 h. Cells were rinsed with ice cold cell-based assay buffer and then incubated with ROS straining buffer. Dihydroethidium (DHE) fluorescence was measured using an excitation wavelength 480 nm and an emission wavelength 580 nm according to manufacturer’s instructions by VICTOR2 (Perkin Elmer, Waltham, MA, USA). Antimycin A was used as positive control [32].

### 2.8. Measurement of Mitochondrial Membrane Potential (△Ψm)

*C. albicans* was added to the wells of a 24-well plate at a density of 5.0 × 10^6^ CFU/mL. NAIMS 7c (1.56, 3.125 and 6.25 μg/mL) was added and further incubated for 2 h. Cells were stained with 5 μM of JC-1 (Biotium, CA, USA) for 30 m at 30 °C. After washing, photographic images were acquired under an inverted fluorescence microscope (EVOS FL Cell Imaging System, Thermo Fisher, Waltham, MA, USA) using the microscope program [17].

### 2.9. Detection for Release of UV Absorbing Materials at 260 and 280 nm

Overnight cultured *C. albicans* was washed with PBS to stop further proliferation. Cells were treated with indicated concentration of NAIMS **7c** for 1 h at 30 °C. Lysate released was measured with the Ultrospec 3000 at 260 and 280 nm. The values were adjusted by subtracting the optical density measured for the corresponding negative control, which was obtained by same compound concentrations in PBS without cells at the same wavelength [33,34].

### 2.10. Quantitative Reverse Transcriptase PCR (qRT-PCR) analysis

qRT-PCR was performed following the method with slight modification [35]. Total RNA was extracted with TRIzol reagent (Invitrogen Corporation, Carlsbad, CA, USA). Real-time PCR was performed in 96-well PCR plates (Bio-rad, Hercules, CA, USA) using 2×RT Pre-MIX kit (Biofact, Daejeon-si, Korea) and CFX Connect Real-Time PCR Detection System (Bio-rad). Primer sequences used in this study are listed in Table 1.

### 2.11. Statistical Analysis

All data are expressed as the mean ± S.D. Significant differences among the groups were determined using the Student’s t-test or one-way ANOVA, and *p* < 0.05 was considered significant.

## 3. Results

### 3.1. Chemistry

First, we tried to synthesize 2-acetyl-1,4-dialkoxynaphthalnes **4**, a key intermediate to target NAIMS **7** by direct alkylation of 1,4-dihydroxynaphthalene followed by Friedel–Crafts acylation. However, Friedel–Crafts acylation was found to be inefficient due to its low yield. Therefore, another synthetic route to produce **4** was suggested as shown in Scheme 1. The compounds **2** were synthesized in high yield from 1,4-diacetoxynaphthalene (**1**) by Fries rearrangement. Then, alkylation of compound **1** with the corresponding alkyl halide afforded compound **3**. The removal of the acetyl group with KOH followed by O-alkylation produced key intermediate **4** in two steps with good yield. Various α-bromination reactions of compound **4** were tested to find a condition optimized for the synthesis of bromoacyl intermediate **5**. Finally, we found that compound **5** was obtained in the best yield when reacted with TBA-Br_3_. *N*-Arylimidazoles such as **6a** and **6d**, which are not commercially available, were synthesized from imidazole and 4-substituted benzyl halide in acetonitrile. The coupling reaction of the 2-bromoacyl intermediates **5** with corresponding imidazoles **6** gave nine new 1,4-dialkoynaphthalen-2-acyl imidazolium salts **7a**–**i**.

Reagents and conditions. (i) BF_3_ 2CH_3_COOH (7 eq), reflux, 1h; (ii) Cs_2_CO_3_ (1.5 eq), R_1_-X (1.5 eq), DMF (0.5M), 70 °C, 2.5 h; (iii) KOH (0.67 eq), MeOH (0.5 M), 0 °C, 1h, then Amberlite IR-120(H), 0 °C, 15 m; (iv) Cs_2_CO_3_ (1.5 eq), R_1_-X (1.5 eq), DMF (0.5M), 70 °C, 1.5 h; (v) TBA-Br_3_ (1.2–1.3 eq), CH_2_Cl_2_ (0.05 M), rt, 3 h; (vi) imidazole (1–2 eq), CH_3_CN, reflux, 1d.

To investigate a simple relationship between structure and activity of NAIMSs, we prepared compounds **8**, **9**, **9′**, NAIMS **10** and NAIMS **11**, as shown in Figure 2. Compounds **8, 9** and **9′** have no imidazolium moiety, and NAIMS **10** and NAIMS **11** have no dialkoxy substituents in a naphthalene ring. The same coupling reaction of readily available 2-bromoacetylnaphthalene or 2-bromoacetyl-1-methoxynaphthalene (**5e**) with imidazole **6b** gave, respectively, NAIMS **10** and NAIMS **11**.

### 3.2. Antifungal Activity of NAIMS **7c** Against Candida spp.

The growth inhibitory activity of **7a–i, 10** and **11** against *Candida* spp. was evaluated using broth microdilution assays. The MIC values of miconazole were used as a positive control and confirmed over 12.5 μg/mL to *Candida* spp. after 24 h incubation. Among the tested compounds, NAIMS **7b**, NAIMS **7c**, NAIMS **7d** and NAIMS **7e** showed stronger antifungal activity against *C. albicans* than miconazole. NAIMS **7c** showed strongest antifungal activity compared with other derivatives; MIC values ranged from 3.125 to 6.25 μg/mL against all *Candida* spp. used in this study. In particular, NAIMS **7c** showed 3.125 μg/mL MIC value in 24 h assay against *C. auris* (KCTC17810) which possessed native resistance to miconazole (Table 2). To check the stability of antifungal activity of synthetic compounds, MIC values were detected every day for 72 h. NAIMS **7c** had 3.125 and 6.25 μg/mL MIC values for *C. albicans* (KACC30071) at 48 and 72 h, respectively. Miconazole, by contrast, showed sustainable anti-*Candida* activity of 25 μg/mL MIC value for 72 h (Table 3). NAIMS **7c** showed stable antifungal activity compared with other derivatives. The yield of NAIMS **7c** was fortunately higher than NAIMS **7b** and NAIMS **7d**, and this suggested that NAIMS **7c** is more cost-beneficial than other derivatives (Scheme 1). In vitro cell viability of NAIMS 7c against HaCaT was performed via MTT assays. NAIMS 7c had IC_50_ of 21.39 μg/mL against HaCaT. Based on results, cell growth test was also evaluated for the serial diluted concentration of NAIMS **7c** in *C. albicans* (KCTC7965, KCTC7270 and KACC30071) for 48 h (Figure 3). Regardless of the time, NAIMS **7c** at 6.25 μg/mL strongly inhibited cell growth compared with miconazole.

### 3.3. Inducing ROS Level and the Loss of Mitochondria Membrane Potential by NAIMS **7c**.

To identify antifungal activity of NAIMS **7c** against *C. albicans*, ROS production in *C. albicans* was detected after NAIMS **7c** treatment. NAIMS **7c** induced higher ROS production than did antimycin. The treatment of 0.78 μg/mL of NAIMS **7c** increased the DHE fluorescence with the maximum at 1.56 μg/mL (Figure 4). For further study about the mechanism of NAIMS **7c**, *C. albicans* was stained with JC-1 dye to detect the mitochondria membrane potential. NAIMS **7c** decreased a red fluorescence at 1.56 μg/mL in *C. albicans*. This suggested that NAIMS **7c** causes damage to mitochondria and alters the cellular state of *C. albicans* (Figure 5). Based on these results, NAIMS **7c** induces loss of mitochondria membrane potential and increases the release of ROS in *C. albicans*.

### 3.4. The Cell Lysis and the Apoptosis of C. albicans by NAIMS **7c**

To determine the induction of the loss of mitochondria membrane potential by NAIMS **7c** treatment, changes in UV absorbing materials were detected by spectrophotometer with NAIMS **7c** in the *C. albicans* culture medium supernatant. The use of 12.5 μg/mL of NAIMS **7c** treatment significantly increased the absorbance at 260 and 280 nm after 1 and 2 h (Figure 6). These results show that treatment of NAIMS **7c** at over the MIC value can lead to lysis of *Candida* spp. and might induce the apoptosis of *Candida* spp. Three different *Candida*-apoptosis related genes were used to detect the effect of NAIMS **7c** in *C. albicans*. YPK1 is a serine/threonine protein kinase that affects diverse cellular activities, including sphingolipid homeostasis. HAC1 is a transcription factor and plays a major role in stress-related transcriptional response. MCA1 is a cysteine protease involved in apoptosis in response to stresses. The expression of these three genes was dramatically increased by NAIMS **7c** treatment (Figure 7).

## 4. Discussion

*Candida* spp. are the most common organism recovered from the skin and blood of hospitalized patients. Although the need to treat them is increasing, the range of antifungal agents available is limited because of their toxicity. In addition, resistant strains and new species that show innate resistance to antifungal agents have been reported [39,40]. Therefore, it is necessary to introduce new antifungal agents to limit the spread of pathogenic fungi.

For the synthesis of 1,4-dialkoy-NAIMS, we developed a highly efficient synthesis method including the synthesis of 1,4-dialkoynaphthalen-2-acyl intermediate **4**, the α-bromination reaction of compound **4**, and the S_N_2 reaction of compound **5** and imidazole **6,** such as a quarternization of imidazole. We found that compound 4 was produced in better yield by a novel synthetic method including Fries-rearrangement of compound 1 followed by alkylation than the direct alkylation of 1,4-dihydroxynaphthalene and subsequent Friedel–Crafts acylation. In addition, it was confirmed that the α-bromination reaction of compound **4** was best under the reaction conditions of TBA-Br_3_.

Among the 14 synthesized compounds (**7a**–**i**, **8**, **9**, **9′**, **10**, **11**), NAIMS **7b**, **7c**, and **7d** showed superior activity compared to other synthetic NAIMSs or miconazole. In particular, NAIMS **7c** is best when considering its antifungal activity, stability and synthesis efficiency. As shown in Figure 8, the energy calculation (Gaussian B3LYP, 6-311 [d, p]) predicts NAIMS **7c** presenting a slightly curved structure similar to phospholipids. Based on these results, we might propose a first-pass reasoning on the relationship between structure and activity as follows. For antifungal activity, the imidazolium ring must be essential. The naphthalene ring requires a 1,4-dialkoxy group, in which the optimal alkyl group is an isoamyl group. Furthermore, it seems better to have no methoxy groups in the naphthalene A ring and no electron withdrawing groups such as NO_2_ in the benzene ring in order for NAIMS to be activated.

Miconazole was used as an antifungal agent which has a fungicidal activity against planktonic *Candida* spp., and the cytotoxic concentration of NAIMS **7c** can be significantly lower than miconazole [41,42]. NAIMS **7c** had IC_50_ of 21.39 and 7.56 μg/mL against HaCaT and *C. albicans* (KACC30071), respectively. Miconazole had IC_50_ of 13.10 and 17.25 μg/mL against HaCaT and *C. albicans*, respectively. Accordingly, the selective index (IC_50_ HaCaT/IC_50_
*C. albicans)* of NAIMS **7c** was higher than miconazole. This result shows that NAIMS **7c** can be more effective to treat *Candida* spp. and safe. Additional in vivo research should be provided to determine the property of any potential candidate for therapeutic applications in the future [43]. Thus, NAIMS **7c** could serve as a promising lead compound for further research. The 1,4-dialkoxy naphthalene-2-acyl compound **9** without the imidazolium moiety and the naphthalene-2-acyl imidazolium **10** without the dialkoxy substituent on the naphthalene ring showed no antifungal activity. However, its hybrid NAIMS **7c** found an excellent antifungal agent. Therefore, as mentioned in the literature, the pharmacophore-hybridization approach is thought to be a useful tool for new drug design and development [44].

NAIMS **7c** led to induced ROS production, the loss of mitochondria membrane potential, the release of UV absorbing materials and up-regulated apoptotic gene expression. ROS and mitochondria play an essential part in apoptosis. These results suggest that NAIMS **7c** induced apoptosis in *C. albicans*.

In summary, a series of novel imidazolium salts containing 2-acetylnaphthalene moiety were designed, prepared and evaluated for antifungal activity. The results presented in this study conclusively demonstrate that NAIMS **7c** has a long-term enhanced antifungal activity against *Candida* spp., including especially *C. albicans* and *C. auris*, which possess native resistance to miconazole, as compared with other compounds including miconazole. Further studies of structure–activity relationships and a wide range of biological activities are ongoing and will be reported in the future.

## Data Availability

The data presented in this study are available on request from the corresponding author.

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
