# Peer review of "Antifungal Activity of 1,4-Dialkoxynaphthalen-2-Acyl Imidazolium Salts by Inducing Apoptosis of Pathogenic Candida spp."

_pharmaceutics, 2021, doi:10.3390/pharmaceutics13030312_

Round 1

Reviewer 1 Report

Review on "Antifungal activity of 1,4-dialkoxynaphthalen-2-acyl imidazo- 2 lium salts by inducing apoptosis of pathogenic Candida spp.

specific comments:

In abstract :

Among the tested compounds, NAIMS 7c showed strongest antifungal activity with 3.125 μg/mL MIC value compared 24 with miconazole which showed 12.5 μg/mL MIC value against Candida spp., and more importantly 25 resistance against C. auris.

Reword, because “resistance” is of the microorganism not form the molecule….

Introduction:

1-Please specify what kind of toxicity the standard of care antifungal exhibited in the introduction section with suitable references. “Drugs currently used against fungal pathogens have limitations because of their toxicity [6, 7]. “ and give information in the manuscript about the toxicity of these new compounds proposed (NMAIS) in vitro for normal cells.

Figures 5: In this figure, we don´t know exactly which one was the control and the sample. “Red fluorescence indicates dye aggregated in the mitochondria, and green indicates dye scattered in the cytoplasm.” It is not obvious, how could you assure that this place is mitochondria. No evidence of this assumption. This figure must be improved in quality and insert a more clear legend. Which mitochondria marker was used to affirm this?

In discussion: “During apoptosis, oxidative stress activates the permeability transition pore (PTP) of mitochondria involved in the release of cytochrome C leading to subsequent metacaspase activation [16].” This assumption in the discussion part (570-572) gives the impression that this occurs upon NAIMS 7c, which is not the case. Please modify this part, considering only data obtained from the manuscript results.

The show clearly the interest of this new molecule, a selectivity index based on experimental data of cytotoxicity in a normal human cell must be provided because as stated the benefit of this NAIMS 7c in terms of safety was not provided in this study. The authors used the argument that the classical standard of care antifungals are toxic to develop new ones..

The discussion should be improved, it should be deepened and the findings better explained based on literature data; relating the MICs of this new molecule to the MICs of the two classes of drugs that were hybridized in the design of proposed new derivatives (imidazole moiety coupled to 1,4 -dialkoxynaphthalen-2acyl).

Author Response

Antifungal activity of 1,4-dialkoxynaphthalen-2-acyl imidazolium salts by inducing apoptosis of pathogenic Candida spp.

We thank the editor and the reviewers for their thoughtful and constructive comments. We have addressed, in a point by point manner, all the suggestions and queries from the reviewer and marked with red in manuscript. The input from the reviewers has allowed us to improve the clarity and quality of our paper. We have included below our point by point response to the reviewers’ comments and have included these additions and alterations to the revised manuscript.

Reviewer 1

specific comments:

In abstract:

“Among the tested compounds, NAIMS 7c showed strongest antifungal activity with 3.125 μg/mL MIC value compared with miconazole which showed 12.5 μg/mL MIC value against Candida spp., and more importantly resistance against C. auris.”

Reword, because “resistance” is of the microorganism not form the molecule….

A: We feel sorry to make you confused. We changed “resistance” to “>100 μg/mL MIC value” to make the sentence naturally.

Introduction:

1-Please specify what kind of toxicity the standard of care antifungal exhibited in the introduction section with suitable references. “Drugs currently used against fungal pathogens have limitations because of their toxicity [6, 7]. “ and give information in the manuscript about the toxicity of these new compounds proposed (NMAIS) in vitro for normal cells.

A: We thank your opinion. We added some references to specify the toxicity. And we also included results of cell viability of NAIMS in vitro with HaCaT.

“In addition, Tuning the toxicity of IMSs as antitumor agents has attracted much attention [12].”

In vitro cell viability of NAIMS 7c against HaCaT was performed via MTT assays. NAIMS 7c had IC50 of 21.39 μg/mL against HaCaT (Data not shown).”

Figures 5:

In this figure, we don´t know exactly which one was the control and the sample. “Red fluorescence indicates dye aggregated in the mitochondria, and green indicates dye scattered in the cytoplasm.” It is not obvious, how could you assure that this place is mitochondria. No evidence of this assumption. This figure must be improved in quality and insert a more clear legend. Which mitochondria marker was used to affirm this?

A: We feel sorry to show the unclear figure and changed it more clearly. This analysis is widely used to detect the mitochondrial membrane potential of cells. We would like to show you several references for this experiment. Please consider our suggestions.

  • Tian H, Qu S, Wang Y, Lu Z, Zhang M, Gan Y, Zhang P, Tian J. Calcium and oxidative stress mediate perillaldehyde-induced apoptosis in Candida albicans. Appl Microbiol Biotechnol. 2017, 101(8):3335-3345.
  • Sun L, Liao K, Hang C, Wang D. Honokiol induces reactive oxygen species-mediated apoptosis in Candida albicans through mitochondrial dysfunction. PLoS One. 2017, 12(2):e0172228.
  • Zhang M, Shi J, Jiang L. Modulation of mitochondrial membrane integrity and ROS formation by high temperature in Saccharomyces cerevisiae. J Biotechnol, 2015, 18.3: 202-209.

In discussion:

“During apoptosis, oxidative stress activates the permeability transition pore (PTP) of mitochondria involved in the release of cytochrome C leading to subsequent metacaspase activation [16].” This assumption in the discussion part (570-572) gives the impression that this occurs upon NAIMS 7c, which is not the case. Please modify this part, considering only data obtained from the manuscript results.

A: We feel sorry for We deleted that sentence to consider only our obtained data as you mentioned.

The show clearly the interest of this new molecule, a selectivity index based on experimental data of cytotoxicity in a normal human cell must be provided because as stated the benefit of this NAIMS 7c in terms of safety was not provided in this study. The authors used the argument that the classical standard of care antifungals are toxic to develop new ones..

A: We appreciate your comment. Following your opinion, we added an in vitro experimental results and its comments in the manuscript.

“The toxicity of NAIMS 7c identified in this study can be significantly lower than the con-centration to be used as an antifungal agent. Additional in vivo research should be provided to determine the property of any potential candidate for therapeutic applications [39].”

The discussion should be improved, it should be deepened and the findings better explained based on literature data; relating the MICs of this new molecule to the MICs of the two classes of drugs that were hybridized in the design of proposed new derivatives (imidazole moiety coupled to 1,4 -dialkoxynaphthalen-2acyl).

A: In discussion part, the antifungal activity of two parts of this new hybrid, 1,4-dialkoxynaphthalen-2-acyl compound 9 and naphthalen-2-acyl imidazolium 10, was further described.

“The 1,4-dialkoxy naphthalene-2-acyl compound 9 without the imidazolium moiety and the naphthalene-2-acyl imidazolium 10 without the dialkoxy substituent on the naphthalene ring did not show good MIC values for antifungal activity. However, their hybrid NAIMS 7c was found to exhibit excellent antifungal activity with low MIC values. Therefore, as mentioned in the literature, the pharmacophore-hybridization approach is thought to be a useful tool for new drug design and development [42].”

Reviewer 2 Report

In this paper, Kim and coauthors focus on the synthesis of 1,4-dialkoxynaphthalen-2-acyl imidazo-2 lium salts (NAIMS) and their antifungal activity toward pathogenic Candida spp. Based on the antifungal activity of N-phenacylimidazolium salts, authors attempt to attach an imidazole moiety to 1,4-dialkoxynaphthalen-2-acyl compound to enhance antifungal activity. The results demonstrated that NAIMS 7c showed superior activity compared to miconazole. Importantly, NAIMS 7c could cause cell busting and the expression of apoptosis-related genes was induced in C. albicans by NAIMS 7c treatment.

  Overall, this is a beautiful work with good writing. It was recommended for publication after minor revision.

  Some questions shown as follows need to be addressed

1. The relevant references such as N-phenacylimidazolium salts and N-phenacylthiazolium for bioactive evaluation should be cited.

2. How about the antifugal activity of 1,4-dialkoxyphenacylimidazolium and 1,4-dialkoxyphenacylthiazolium.

Author Response

Thank you, sincerely.

Reviewer 3 Report

Candida spp. is the most common fungal pathogen. There are different available options to treat this infection, however the arise of resistance requires the development of new compounds. In this manuscript, the authors report the synthesis of a new compound to treat fungal infections caused by Candida spp. The experiments are well described and executed, however there are some comments that would result useful to improve the quality of this manuscript.

  • Abstract: there’s a typo. It should be Candida spp instead of Candida sp and Candida should always appear in capital letters. Please correct.
  • Introduction: what’s an AGE blocker? Please specify.
  • 2.4.1: MC stands for dichlorometane? I think using MC sounds a bit confusing, I suggest the authors would use DCM instead as it’s more frequently used.
  • Table 1: as all strains have been purchased from KTTC or KACC, I suggest this would appear in the text rather than in the table. The authors could also specify KTTC or KACC for each strain instead of giving a general statement.
  • 7: the whole section appears in italics. I guess this must have been an edition error, but just in case.
  • I’d suggest a change in the headers in section 3. For example, “antifungal activity” rather than “NAIMS 7c possessed strongest antifungal activity against Candida spp.”. This statement looks fine, but I think it should be included in the text rather than the header.
  • Table 3 appears twice.
  • Table 4 looks very large and it’s quite difficult to follow. I’d suggest the authors would use a histogram instead.
  • Figures 3-7: legend needs to be improved. It’d make easier to understand what the figures are showing.

Author Response

Antifungal activity of 1,4-dialkoxynaphthalen-2-acyl imidazolium salts by inducing apoptosis of pathogenic Candida spp.

We thank the editor and the reviewers for their thoughtful and constructive comments. We have addressed, in a point by point manner, all the suggestions and queries from the reviewer and marked with red in manuscript. The input from the reviewers has allowed us to improve the clarity and quality of our paper. We have included below our point by point response to the reviewers’ comments and have included these additions and alterations to the revised manuscript.

Reviewer 3

Candida spp. is the most common fungal pathogen. There are different available options to treat this infection, however the arise of resistance requires the development of new compounds. In this manuscript, the authors report the synthesis of a new compound to treat fungal infections caused by Candida spp. The experiments are well described and executed, however there are some comments that would result useful to improve the quality of this manuscript.

Abstract: there’s a typo. It should be Candida spp instead of Candida sp and Candida should always appear in capital letters. Please correct.

A: We appreciate your comments. And we changed all the incorrect letters to be correct.

Introduction: what’s an AGE blocker? Please specify.

A: We feel sorry to make you confused. In order to specify an AGE blocker, we changed ‘AGE blocker’ to ‘advanced glycation end-products (AGEs) breaker reversing one of the main mechanisms of ageing’.

2.4.1: MC stands for dichlorometane? I think using MC sounds a bit confusing, I suggest the authors would use DCM instead as it’s more frequently used.

A: We thank your suggestion. We changed the abbreviation for dichloromethane from ‘MC’ to ‘DCM’ at your suggestion.

Table 1: as all strains have been purchased from KTTC or KACC, I suggest this would appear in the text rather than in the table. The authors could also specify KTTC or KACC for each strain instead of giving a general statement.

A: We thank your comments and moved the purchasing information to the text, section 2.3. And we deleted the table 1.

7: the whole section appears in italics. I guess this must have been an edition error, but just in case.

A: We thank your mention and checked the recommended format of this journal and each section’s title should be in italics.

I’d suggest a change in the headers in section 3. For example, “antifungal activity” rather than “NAIMS 7c possessed strongest antifungal activity against Candida spp.”. This statement looks fine, but I think it should be included in the text rather than the header.

A: We appreciate your suggestion. As yours, we changed the result titles shortly, but implicationally.

Table 3 appears twice.

A: We feel sorry for that mistake sincerely. We deleted the replicate data.

Table 4 looks very large and it’s quite difficult to follow. I’d suggest the authors would use a histogram instead.

A: Following your considerate suggestion, we changed the table 4.

Figures 3-7: legend needs to be improved. It’d make easier to understand what the figures are showing.

A: We appreciate your opinion. We changed our legend slightly to understand easily.

“Figure 3. The growth of C. albicans was inhibited by NAIMS 7c. Microdilution assay was per-formed at 3.125 μg/mL for 48 h. Data represent mean ± SD from three independent experiments. **p<0.05 and ***p < 0.01.”

“Figure 5. NAIMS 7c changed the mitochondrial membrane potential in C. albicans. 1.56 μg/mL of NAIMS 7c was treated for 2 h. Cells were detected by fluorescence microscope. Red fluorescence indicates dye aggregated in the mitochondria, and green indicates dye scattered in the cytoplasm.”

“Figure 6. NAIMS 7c induced the cell lysis of C. albicans. The release-detecting assay was per-formed after 1 or 2 h. Data represent mean ± SD from three independent experiments. **p < 0.05.”

Round 2

Reviewer 1 Report

Please, observe this modified phrase in the revised version “The toxicity of NAIMS 7c identified in this study can be significantly lower than the concentration to be used as an antifungal agent. "

Toxicity is a word related to in vivo studies, this is not the case for this work.

This assumption seems to be the opposite as you have found with the HaCat results of cytotoxicity. Please reword correctly. Suggestion: "The cytotoxic concentrations instead of “The toxicity of NAIMS 7c identified in this study can be significantly lower than the concentration to be used as an antifungal agent.

Please to be clear, calculate the selectivity index based on IC50 HaCat/IC50 Candida sp. for  NAIMS 7c and include this in the discussion to improve the quality of the discussion with numeric data. Compare in the discussion with the selective index of other antifungals used to treat Candida species used in this work, fluconazole, caspofungin, for example. Miconazole is not good positive control, there are newer effective antifungals.

Author Response

Thank you, sincerely.
